# Adversarial Text Generation by Search and Learning

**Guoyi Li[1], Bingkang Shi[1], Zongzhen Liu[1], Dehan Kong[2], Yulei Wu[3],**
**Xiaodan Zhang[1], Longtao Huang[2], Honglei Lyu[1]**

[1] Chinese Academy of Sciences, Institute of Information Engineering, Beijing, China
[2] Alibaba Group, Alibaba Artificial Intelligence Governance Laboratory, Beijing, China
[3] University of Bristol, Department of Electrical and Electronic Engineering, Bristol, UK
{liguoyi, shibingkang, zhangxiaodan}@iie.ac.cn

## Abstract

Recent research has shown that evaluating the robustness of natural language processing models using textual attack methods is significant. However, most existing text attack methods only use heuristic replacement strategies or language models to generate replacement words at the word level. The blind pursuit of high attack success rates makes it difficult to ensure the quality of the generated adversarial text. As a result, adversarial text is often difficult for humans to understand. In fact, many methods that perform well in terms of text attacks often generate adversarial text with poor quality. To address this important gap, our work treats black-box text attack as an unsupervised text generation problem and proposes a search and learning framework for Adversarial Text Generation by Search and Learning (ATGSL) and develops three adversarial attack methods (ATGSL-SA, ATGSL-BM, ATGSL-FUSION) for black-box text attacks. We first apply a heuristic search attack algorithm (ATGSL-SA) and a linguistic thesaurus to generate adversarial samples with high semantic similarity. After this process, we train a conditional generative model to learn from the search results while smoothing out search noise. Moreover, we design an efficient ATGSL-BM attack algorithm based on the text generator. Furthermore, we propose a hybrid attack method (ATGSL-FUSION) that integrates the advantages of ATGSL-SA and ATGSL-BM to enhance attack effectiveness. Our proposed attack algorithms are significantly superior to the most advanced methods in terms of attack efficiency and adversarial text quality.

## 1 Introduction

Recent research has demonstrated that deep neural networks (DNNs) are vulnerable to maliciously crafted adversarial text examples that can fool victim models into making wrong predictions (Wang, 2018; Papernot et al., 2016a,b). These text examples that add malicious perturbation to the original text do not affect human judgment but can deceive deep learning models. (Bender and Koller, 2020) pointed out that deep neutral models have defects in understanding the meaning conveyed by language. Thus, generating adversarial samples has become a common method for evaluating the weakness and robustness of DNNs.

Existing malicious text generation algorithms can be classified into character-level attacks, sentence-level attacks, and word-level attacks. *Character-level attacks* (Belinkov and Bisk, 2017; Ebrahimi et al., 2017) include the addition, deletion, replacement, and order exchange of characters, which sacrifices the readability of the generated text in exchange for the attack's success rate. *Sentence-level attacks* (Jia and Liang, 2017; Inui et al., 2019) regard the original input of the whole sentence as a perturbation object, which often makes a considerable difference between the generated text and the original input. It is not easy to guarantee the quality of the generated text. Therefore, many studies focus on improving the attack success rate and the quality of the attack text generated by word replacement, hence *word-level attacks*.

Previous works mainly generate textual attacks on word replacement according to specific rules (Alzantot et al., 2018; Ren et al., 2019; Jin et al., 2020; Bender and Koller, 2020; Zang et al., 2020b; Yang et al., 2021a). Most of these models show good attack performance by using multiple linguistic constraints (e.g., NER tagging and POS tagging) and a well-organized linguistic thesaurus (e.g., WordNet and HowNet). However, these works require extensive preprocessing, and the substitution words selection heavily relies on tags and cannot guarantee the fluency and grammaticality of adversarial samples. The other attack methods based on language models, such as BERT, can generate contextual perturbations (Garg and Ramakrishnan, 2020; Li et al., 2020b; Yang et al., 2021b;

Malik et al., 2021). These models ensure the predicted token fits the sentence well but cannot preserve the semantic similarity (Yang et al., 2021a). For example, in the sentence "I feel [MASK]", predicting the [MASK] as happy or sad is equally sensible but results in a sentiment analysis task. In order to improve these issues, recent research has focused on learning-based methods (Zang et al., 2020a; Lee et al., 2022; Sabir et al., 2021), aiming to utilize model learning to improve the balance between the efficiency of attack algorithms and the quality of adversarial texts from the attack evaluation history.

Inspired by these works, adversarial text generation can be seen as an unsupervised text generation problem. Thus, we propose a new framework for Adversarial Text Generation by Search and Learning (ATGSL). This framework includes a search module that uses strong search algorithms (e.g., Simulated Annealing) to search synonym spaces and a learning module (e.g., BERT-MLM) that learns from search results. We first use the Simulated Annealing (SA) optimization algorithm in each step to determine token replacement priority. Then we accept or reject suggestions based on a heuristic-defined scoring function and save successful attack adversarial samples (ATGSL-SA). Since ATGSL-SA requires a large number of iterations and attack effectiveness is easily affected by initial conditions, we utilize search results as pseudo references for training condition generators. We design an efficient ATGSL-BM attack algorithm based on generators. In addition, we propose a hybrid attack method (ATGSL-FUSION) to enhance attack effectiveness.

- We propose a black-box attack framework based on a search and learning framework to improve the balance of the attack efficiency and the quality of adversarial samples. To the best of our knowledge, this is the first of its kind to propose search and learning methods to generate adversarial samples.

- ATGSL-SA generates word substitutions from both synonym candidates and sememe candidates. It integrates label score, replacement word rate, and semantic similarity into the design of the SA algorithm to generate adversarial texts with higher semantic similarity.

- Our attack method ATGSL-BM based on fine-tuned pre-trained language models, improves

attack effectiveness and the quality of adversarial texts. In addition, ATGSL-FUSION has the best attack success rate because it improves the impact of the initial conditions on the ATGSL-SA search.

- Extensive experimental results show that our model significantly improves the existing state-of-the-art models in adversarial text generation in terms of attack efficiency and text quality.

## 2 Related Work

### 2.1 Textual Adversarial Attack

Existing textual adversarial attack methods can mainly be classified into character-level, sentence-level, and word-level attacks based on the granularity of perturbations. Character-level attacks mainly operate on the addition, deletion, substitution, and exchange order of characters in the original input, and typical substitution methods such as random substitution (Belinkov and Bisk, 2017), character similarity substitution (Eger et al., 2019), etc. However, character-level attacks often produce low-quality adversarial samples that can violate grammar rules and be resisted by grammar-based defence methods (Pruthi et al., 2019). Sentence-level attacks treat the original input of the whole sentence as an object of perturbation. Typical examples of such attacks include paraphrasing (Ribeiro et al., 2018; Iyyer et al., 2018), encoding-decoding (Zhao et al., 2017), but the generated text can cause a significant discrepancy with the original text. Word-level attacks perturb words in the original input, and word replacement is the primary method. Common word replacement methods can be categorized into rule-based attacks and learning-based attacks.

Rule-based attacks mainly select candidate words based on the preset strategy. (Jin et al., 2020) developed methods to search for replacement words relying on cosine distance to find adjacent word vectors in Glove's embedding space, which may lead to the opposite meaning of original words. To avoid this limitation, (Ren et al., 2019) selected candidate words from a well-organized linguistic thesaurus (e.g., WordNet (Miller, 1998) and HowNet (Dong and Dong, 2006)) and chose appropriate replacement words with the optimal strategy. However, this work focuses on a fixed WIS (Weight Important Score) order, leading to local selection and excessive word replacement. To reduce

the effect of static WIS on word selection order, BESA (Yang et al., 2021b) optimized the step size of Simulated Annealing to choose the best token replacement combination. However, this method is easily influenced by the initial condition and generates many queries for candidate sets. Recent language model-based attacks are mainly based on BERT Masked Language Model (BERT-MLM). (Garg and Ramakrishnan, 2020; Li et al., 2020b) used BERT-MLM to score replacement words. Although these methods consider the semantic relevance of context, they still cause ambiguity in tasks such as rumor detection and emotion analysis. This is because the candidate words generated by these pre-trained models do not consider the role of the original words that were replaced by [MASK].

Learning-based attacks learn from evaluation history to improve learning model parameters (Zang et al., 2020a; Lee et al., 2022; Sabir et al., 2021). (Zang et al., 2020b) utilized the probability feedback of the target model to modify the parameters of particle swarm optimization (PSO) in order to select the replacement better. BBA (Lee et al., 2022) utilized Bayesian algorithms to learn query history and optimized the selection of replacement locations. But these methods of fitting the target model in the evaluation history have high computational complexity and uncertainty. On the other hand, ReinforceBug (Sabir et al., 2021) designed a feedback generation method including grammar evaluation based on a reinforcement learning framework, which improves the quality of the generated samples, but also has the drawbacks of the above WIS score-based method, which can affect attack performance. Inspired by this, our work aims to improve the balance between attack efficiency and the quality of adversarial texts by attempting to build the framework by virtue of both search and learning.

## 2.2 Unsupervised Text Generation

Neural unsupervised text generation has made significant progress, with variational autoencoders (Kingma and Welling, 2013) being a well-known approach. Search-based methods have also been developed for various text generation tasks (Kumar et al., 2020; Schumann et al., 2020; Miao et al., 2019; Liu et al., 2019a), but they are not learnable. (Li et al., 2020a) proposed a search-and-learning approach to improve performance and inference efficiency. Our paper adopts this approach but differs

in several ways: 1) Our search aims to obtain adversarial texts with higher semantic similarity and quality; 2) We use search and learning to design three attack algorithms: ATGSL-SA, ATGSL-BM, and ATGSL-FUSION. To the best of our knowledge, we are the first to address adversarial text generation using the search-and-learning method.

## 3 Problem Statement

This paper focuses on black-box attacks, in which attackers do not know the internal structure and parameters of the target model, and can only query the target model to obtain its output relative to a given input. In this study, attackers can query the target model's output labels and confidence scores.

Formally, let $\mathbf{X} = \{x_1, x_2, ..., x_n\}$ be the input dataset including $n$ samples, and each $x^{(i)}$ corresponds to a ground-truth label $y_i^{true} \in \mathbf{Y}$. Let $\mathbf{F} : \mathbf{X} \to \mathbf{Y}$ be a well-trained model that classifies input samples into labels. *This attack can generally be modelled as an optimization problem, which aims to mislead the target model by the adversarial sample $\mathbf{X}_{adv}$ with better quality of adversarial texts:*

$$\mathbf{F}(\mathbf{X_{adv}}) \neq \mathbf{Y}. \tag{1}$$

An adversarial text example $\mathbf{X}_{adv}$ is defined as the original input $\mathbf{X}$ that has been subjected to slight perturbations $\Delta\mathbf{X}$, i.e., $\mathbf{X}_{adv} = \mathbf{X} + \Delta\mathbf{X}$.

## 4 Methodology

Our attack algorithms are summarized in Algorithm 1. In our approach, we first employ the ATGSL-SA algorithm to search for suitable replacement words in synonym and sememe word lists, ensuring semantic similarity while identifying appropriate replacement word combinations for attacks. To address the issue of high iteration time cost and the tendency to get trapped in local optima due to its initial condition in SA (Henderson et al., 2003), we designed ATGSL-BM and ATGSL-FUSION [1]. In ATGSL-BM, we fine-tune a pre-trained language model (LM) to learn the search patterns of the ATGSL-SA algorithm and serve as the foundation for the attack algorithm. The large model capacity and extensive pre-training of LMs enable the generation of high-quality adversarial texts. In ATGSL-FUSION, we generate intermediate solutions through ATGSL-BM to improve the sensitivity of the initial condition in ATGSL-SA.

---

[1] https://github.com/DABAI6666/ATGSL

**Algorithm 1:** Our Proposed Algorithms

---

**Input:** Original text $\mathbf{X}$, initial search state $\mathbf{X}_{ini}$, target model $\mathbf{F}$, well-trained conditional generative model $\mathcal{B}$

**Output:** Adversarial sample $\mathbf{X}_{adv}$

1  **Initialization**: The initial temperature $T_0 = T_{init} = 0.1$, internal simulation steps $\text{MaxStep} = 20$, $\mathbf{S}$ is initially an empty set, the initial adversarial example $\mathbf{X}_{adv} = \mathbf{X}_{ini}$

2  **for** *each token $w_i \subset \mathbf{X}$* **do**

3     Candidate set $\mathbf{C}_i$ sampled from synonym space $\mathbb{W}$ (WordNet) and sememe space $\mathbb{H}$ (HowNet);

4  **for** $t = 1, \cdots, \text{MaxStep}$ **do**

5     **if** *method == ATGSL-SA or ATGSL-FUSION* **then**

6         Randomly choose an edit position $k$ and utilize $C_k$ to replace $x_k$ in Eq. 3 to craft $\mathbf{X}_{new}$;

7         Compute the objective value $s(*)$ by Eq. 4;

8     **if** *method == ATGSL-BM* **then**

9         Randomly choose an edit position $k$ to replace position set and use the well-trained generative model $\mathcal{B}$ to generate a new word combination as $\mathbf{X}_{new}$ in Eq. 3;

10     **if** $s(\mathbf{Y}|\mathbf{X}_{new}) - s(\mathbf{Y}|\mathbf{X}_{adv}) < 0$ **then**

11         $\mathbf{X}_{adv} = \mathbf{X}_{new}$;

12     **else**

13         Compute the probability $p'$ by Eq. 9;

14         **if** *method == ATGSL-SA* **then**

15             With probability $p$, $\mathbf{X}_{adv} = \mathbf{X}_{new}$;

16         **if** *method == ATGSL-BM* **then**

17             With probability $1$-$p$, discard edit position $k$;

18         **if** *method == ATGSL-FUSION* **then**

19             With probability $p$, discard the replaced $\mathbf{X}_{adv}$ and utilize $\mathcal{B}$ to generate new $\mathbf{X}_{adv}$ to change the initial condition;

20     Calculate $T_0'$ by Eq. 10, $T_0 = max\{T_0', 0.01\}$;

21     **if** $\mathbf{F}(\mathbf{X}_{adv}) \neq \mathbf{Y}$ **then**

        **Return:** $\mathbf{X}_{adv}$

    **Return:** Adversarial sample $\mathbf{X}_{adv}$

22

---

## 4.1 The ATGSL-SA Algorithm

In ATGSL-SA, we regard determining the best word replacement order as a combinatorial optimization problem, and use SA to decide word replacement priority. Simulated Annealing (SA) is an effective heuristic search algorithm, suitable for searching large discrete or continuous spaces (Kirkpatrick et al., 1983; Granville et al., 1994). Therefore, we choose SA as our search text attack algorithm to search synonym and sememe space and generate adversarial samples with high semantic similarity.

**Candidate Word List.** In a sentence composed of $m$ words $\mathbf{X} = \{w_i, w_2, ..., w_m\}$, only some keywords have an impact on the prediction model $F$.

This is consistent with the research by (Niven and Kao, 2019), who found that the language model focuses on statistical clues of some words. Therefore, we preprocess all candidate words of each token and design a priority mechanism to select the best replacement word. Since the attack method based on linguistic thesaurus has been proved to have a higher semantic similarity between the generated text and the original text (Zang et al., 2020b; Yang et al., 2021a), our work gets initial candidate words from synonym-based and sememe-based substitution $\mathbf{C}_i = \mathbb{W} \cup \mathbb{H}$, as shown in lines 3~4 of Algorithm 1. For each potential candidate $w_i' \in \mathbb{C}_i$ who replaces the original word $w_i$, we define $\mathbf{X}_{w_i}' = w_1, w_2, ..., w_i', ..., w_m$ and candidate importance score $I_{w_i'}$ as the true score probability reduction:

$$I_{w_i'} = P(\mathbf{Y}_{true} \mid \mathbf{X}) - P(\mathbf{Y}_{true} \mid \mathbf{X}_{w_i}'). \quad (2)$$

In every search for replacement words, we choose the word $w_i$ with the highest $I_{w_i}'$ as the best replacement word $w_i^*$. The synonym candidate selection function is given below:

$$w_i^* = R(w_i, \mathbf{C}_i) = \underset{w_i' \in \mathbb{C}_i}{\arg\max}\, I_{w_i'} \quad (3)$$

**Search Process.** In the search process, we regard each molecule in the search space corresponds to a word in the sentence, and the replacement of each word corresponds to the movement of the molecular position. During the simulation process, any molecular motion that can reduce the objective function will be accepted, and there is also a conversion probability of increasing the objective function. Additionally, this approach can reduce the impact of word replacement order because sometimes replacing two words $\{top1, top3\}$ can be even better than changing the top-3 WIS words $\{top1, top2, top3\}$.

Our goal in the ATGSL-SA is to find new adversarial samples that can minimize the probability of true labels while maintaining high semantic similarity. The heuristic-based objective function in line 11 of Algorithm 1 is:

$$s(\mathbf{Y}|\mathbf{X}_{new}) = P_{true}(\mathbf{X}_{new}) + \alpha \times Dis(\mathbf{X}, \mathbf{X}_{new}) + \beta \times (1 - Sem(\mathbf{X}, \mathbf{X}_{new})),$$
$$(4)$$

where $Dis(\mathbf{X}, \mathbf{X}_{new})$ represents the number of different words between $\mathbf{X}$ and $\mathbf{X}_{new}$, $Sem(\mathbf{X}, \mathbf{X}_{new}) \in [0, 1]$ (the higher, the better) is calculated by Universal Sense Encoder (USE)

(Cer et al., 2018), and $\alpha$, $\beta$ are parameters to make a tradeoff between the attack efficiency and the semantic similarity. In our implementation, we empirically set $\{\alpha, \beta\} = \{0.01, 0.1\}$. Please refer to Appendix A for more details of ATGSL-SA.

## 4.2 The ATGSL-BM Algorithm.

The ATGSL-BM algorithm consists of two stages: training and attack. During the training stage, the text conditional generator learns the attack patterns of ATGSL-SA. The training procedure is summarized in Algorithm 2. During the attack stage, we use the trained conditional generator to generate text candidates. Fig. 2 and Algorithm 2 in Appendix B provides a diagram of the training stage.

**Training Process.** The local search algorithm has low computational efficiency during inference, requiring several hundred steps of editing and re-evaluation for each sample. Due to the state-of-the-art performance of the encoder-decoder framework of BERT-MLM (BM) (Vaswani et al., 2017) in text generation, our intuition is to fine-tune BERT-MLM based on ATGSL-SA's search results. On the other hand, BERT-MLM provides a new way of selecting candidate words, which utilizes a bidirectional language model to determine two candidate words $\{top1, top2\}$ through context, unlike ATGSL-SA, which determines $top1$ and then calculates $P(top2|top1)$. This method further reduces the impact of word replacement orders. Specifically, we use [MASK] to fill in the replacement word for $\mathbf{X}_{adv}$ obtained from ATGSL-SA to obtain the $\mathbf{X}^{mask} = \{x_0, ..., x_i^{mask}, ..., x_n\}$. In the training process, we construct question-answer pairs $(\mathbf{X}, \mathbf{X}^{mask})$ as input for the encoder and $(\mathbf{X}, \mathbf{X}_{adv})$ as the label.

$$h_i = E_{in}(x_i), \tag{5}$$

$$(\tilde{h}_0^{mask}, ..., \tilde{h}_i^{mask}, ..., \tilde{h}_n^{mask})$$
$$= f_{dec}(f_{enc}(h_0, ..., h_i^{mask}, ..., h_n)), \tag{6}$$

$$\tilde{x}_i = argmax(softmax(E_{out}(\tilde{h}_i^{mask}))), \tag{7}$$

where $f_{enc}$ and $f_{dec}$ are the encoder and decoder. Given a source sequence $x$, the objective is the word-by-word cross-entropy (CE) loss, given by

$$J_{CE} = -\sum_{n=1}^{N} \sum_{v \in V} y_{i,v}^{(SA)} log\, p_{i,v}^{(BM)}, \tag{8}$$

where $y_{i,v}^{(SA)}$ is a binary value, indicating whether the $i$th [MASK] is $v$ or not in the ATGSL-SA's output for this data sample, and $log\, p_{i,v}^{(BM)} = Pr[y_i = v|y^{(SA)}, x^{mask}]$, which the BERT-MLM predicts.

Table 1: Statistics of the datasets

| Task | Dataset | Train | Test | Classes | Avg Len |
|---|---|---|---|---|---|
| Classification | AG'News | 27K | 9K | 4 | 43 |
| | IMDB | 25K | 25K | 2 | 227 |
| | MR | 7K | 3K | 2 | 30 |
| Entailment | MNLI | 430k | 10K | 3 | 15 |
| | SNLI | 560k | 10K | 3 | 12 |

In short, due to the reduced cross-entropy loss, Eq. 8 is equivalent to reducing $KL(\hat{y}^{(SA)} \| p_i^{(BM)})$. Minimizing the KL-term makes the slot $p_i^{(BM)}$ more wide-spreading than the $\hat{y}^{(SA)}$ because of asymmetry nature, which explains why CE-trained BERT-MLM can smooth out the noise of the stochastic SA search (Li et al., 2020a).

**Attack Process.** Our following insight is to use trained BM to design an ATGSL-BM algorithm for generating adversarial texts. Similar to ATGSL-SA search for adversarial texts, in each round, this algorithm determines the new replacement position randomly in line 10 of Algorithm 1. We mask out $m$ corresponding positions and obtain the top-t candidate words for each replacement position from the trained BERT-MLM, we list all possible candidates sentence $S \in t \times m$. Which is $m^t$ candidate. We use to calculate the semantic similarity between all candidates and original text and select top-k candidates. And choose the one with the best attack effect as $\mathbf{X}_{new}$.

## 4.3 The ATGSL-FUSION Algorithm.

ATGSL-FUSION is a hybrid algorithm of ATGSL-SA and ATGSL-BM. For instance, in each step of ATGSL-SA, a set of positions to be replaced is selected during the initial stage. After selecting several replacement words $\{w_1, w_2, w_3\}$, if multiple attack attempts $\mathbf{X}_{new}$ do not decrease score $s(Y_{true}|X_{new})$, this local optima problem caused by the initial conditions will waste considerable computational time. In ATGSL-FUSION, by altering the initial conditions and using the results generated by ATGSL-BM as an intermediate solution $\mathbf{X}_{new}$ for further search, we can avoid local optima in line 20 of Algorithm 1. And proceed with the next iteration based on the modified $\mathbf{X}_{adv}$ in the space of synonyms and sememe.

## 5 Experiments

## 5.1 Experiment Settings

**Datasets.** We evaluate the proposed ATGSL and its variants on five public datasets, including IMDB

Table 2: The attack success rate (ASR) of various attack algorithms on text datasets.

| Dataset | Model | PWWS | TFEO | PSO | Reinforce-Bug | BEAT | BBA | BESA | ATGSL-SA | ATGSL-BM | ATGSL-FUSION |
|---------|-------|------|------|-----|---------------|------|-----|------|----------|----------|--------------|
| MR | CNN | 91.8% | 92.1% | 93.1% | 90.7% | 93.1% | 93.8% | 95.1% | 96.8% | 97.9% | **99.5%** |
| | LSTM | 89.4% | 90.1% | 91.3% | 88.7% | 92.8% | 92.6% | 94.2% | 95.7% | 98.2% | **99.6%** |
| | BERT | 85.7% | 83.9% | 88.4% | 81.6% | 82.8% | 92.8% | 93.2% | 94.1% | 97.4% | **98.9%** |
| | RoBERTa | 82.8% | 83.2% | 87.2% | 79.7% | 81.2% | 91.5% | 90.8% | 92.3% | 97.3% | **98.7%** |
| IMDB | CNN | 94.1% | 96.6% | 98.5% | 96.7% | 98.2% | 98.4% | 98.4% | 98.7% | 96.4% | **99.5%** |
| | LSTM | 94.3% | 95.8% | 97.6% | 92.2% | 96.4% | 95.5% | 97.3% | 97.8% | 95.4% | **98.9%** |
| | BERT | 77.8% | 75.2% | — | 83.9% | 89.6% | 88.5% | 93.3% | 95.4% | 94.3% | **98.5%** |
| | RoBERTa | 74.2% | 78.9% | — | 82.1% | 85.6% | 86.8% | 92.4% | 94.2% | 93.5% | **97.6%** |
| AG's News | CNN | 82.3% | 81.7% | 83.9% | 81.5% | 88.4% | 90.2% | 88.6% | 91.6% | 91.9% | **93.2%** |
| | LSTM | 78.6% | 78.7% | 80.8% | 77.7% | 85.8% | 86.4% | 84.3% | 91.8% | 92.3% | **94.1%** |
| | BERT | 73.6% | 73.2% | 77.8% | 74.8% | 83.3% | 82.7% | 86.3% | 88.5% | 89.3% | **92.8%** |
| | RoBERTa | 72.5% | 73.5% | 81.3% | 79.8% | 83.6% | 81.5% | 85.2% | 87.8% | 88.4% | **93.3%** |
| MNLI | InferSent | 85.7% | 85.2% | 86.6% | 84.3% | 87.8% | 90.6% | 91.5% | 92.6% | 94.7% | **97.5%** |
| | ESIM | 80.3% | 82.6% | 83.5% | 81.5% | 86.3% | 85.6% | 85.7% | 87.6% | 90.3% | **92.4%** |
| | BERT | 82.1% | 81.9% | 82.8% | 78.4% | 84.7% | 85.4% | 84.4% | 86.3% | 92.4% | **96.7%** |
| | RoBERTa | 80.2% | 81.4% | 81.9% | 80.4% | 83.1% | 84.5% | 83.6% | 84.7% | 92.9% | **95.2%** |
| SNLI | InferSent | 91.7% | 92.2% | 93.8% | 90.5% | 95.8% | 95.6% | 96.8% | 97.9% | 98.4% | **99.2%** |
| | ESIM | 88.3% | 87.6% | 89.5% | 85.5% | 90.3% | 90.2% | 90.7% | 91.5% | 93.2% | **95.3%** |
| | BERT | 90.3% | 89.8% | 92.3% | 90.4% | 94.3% | 92.7% | 93.5% | 95.7% | 96.8% | **98.6%** |
| | RoBERTa | 88.9% | 89.4% | 91.5% | 89.4% | 93.1% | 91.9% | 92.8% | 94.7% | 97.3% | **98.9%** |

(Maas et al., 2011), MR (Pang and Lee, 2005), AG's News (Zhang et al., 2015), MNLI matched (Williams et al., 2017) and SNLI (Bowman et al., 2015). The AG's News, IMDB, and MR are used for classification tasks, whereas MNLI and SNLI are used for textual entailment. Statistical details of these datasets are summarized in Table 1.

**Target models.** We apply our attack algorithm to popular target models, i.e., CNN (Kim, 2014), LSTM (Hochreiter and Schmidhuber, 1997), BERT (Devlin et al., 2018) and RoBERTa (Liu et al., 2019b) on sentence classification tasks, and the standard InferSent (Conneau et al., 2017), ESIM (Chen et al., 2016), BERT and RoBERTa on textual entailment tasks. CNN is stacked by a word embedding layer with 50 embedding dimensions, a convolutional layer with 250 filters, and each kernel size of 3. LSTM passes the input sequence through a 100-dimension embedding layer, concatenating a 128-unit long short-term memory layer, and following a dropout of 0.5. We download BERT (bert-base-uncased), RoBERTa (roberta-base) from the Transformers model hub HuggingFace[2]. The original test results are listed in Table 7. Please refer to Table 7 in Appendix D.1 to obtain the test success rates of each model on datasets.

**Baselines.** We compare our method with these baselines such as PWWS, TFEO, PSO, Reinforce-Bug, BEAT, BESA. PWWS, TFEO, BEAT are rule-based attacks and the PSO and BESA are Learning-based attacks (Ren et al., 2019; Jin et al., 2020; Zang et al., 2020b; Sabir et al., 2021; Li et al., 2020b; Yang et al., 2021b; Lee et al., 2022). Please

refer to Appendix C.2 for more details of these attacks.

For all datasets, we evaluate the attack success rate, average word substitution rate, semantic similarity score, grammatical Errors score and cost time as shown in Tables 2, 3 and 4. ASR is defined as the misclassification rate of the target model. In our experiment, semantic similarity and grammar (grammatical errors) are calculated by Universal Sense Encoder (USE) [3] and LanguageTool [4], respectively.

## 5.2 Experimental Results

**Main Results.** Tables 2 and 3 show the performance of our proposed method, ATGSL, and all compared methods on five datasets. The results demonstrate that ATGSL outperforms all state-of-the-art models. Compared to fixed WIS algorithms such as PWWS and TEFO, other algorithms such as BESA that use SA to optimize token replacement combinations have better attack performance because they consider the impact of substitution word selection order on WIS. Although PSO utilizes heuristic methods to optimize substitution word selection, it takes too much time to attack very deep models (BERT, RoBERTa) with long text input (such as IMDB). While heuristic-based strong search algorithms have significant effects, a large number of iterations can result in expensive attack costs. On the other hand, algorithms based on language models, such as BEAT and BESA, have fewer grammatical errors but cause redundancy of

[2]https://huggingface.co/models

[3]https://tfhub.dev/google/ universal-sentence-encoder
[4]https://languagetool.org

Table 3: Automatic evaluation results of adversarial example quality. "%M", "%I" and "%S" indicate the modification rate, the semantic similarity, grammatical error increase rate, respectively.

| Method | Dataset | MR | | | IMDB | | | AG's News | | |
|---|---|---|---|---|---|---|---|---|---|---|
| | | %M | %I | %S | %M | %I | %S | %M | %I | %S |
| PWWS | CNN | 13.1 | 7.4 | 0.69 | 1.8 | 3.5 | 0.87 | 6.3 | 7.8 | 0.72 |
| | LSTM | 13.5 | 8.3 | 0.67 | 2.1 | 3.3 | 0.89 | 8.1 | 8.4 | 0.63 |
| | BERT | 14.5 | 9.9 | 0.63 | 5.2 | 4.3 | 0.81 | 10.3 | 9.6 | 0.57 |
| | RoBERTa | 15.2 | 10.5 | 0.58 | 5.8 | 4.2 | 0.80 | 11.2 | 10.1 | 0.55 |
| TEFO | CNN | 17.3 | 9.4 | 0.73 | 2.8 | 3.2 | 0.84 | 7.5 | 7.3 | 0.69 |
| | LSTM | 15.4 | 8.7 | 0.68 | 3.1 | 3.1 | 0.83 | 8.6 | 7.6 | 0.64 |
| | BERT | 20.2 | 10.9 | 0.63 | 6.3 | 3.6 | 0.78 | 9.3 | 8.4 | 0.54 |
| | RoBERTa | 21.3 | 11.4 | 0.62 | 6.4 | 3.8 | 0.78 | 9.8 | 8.3 | 0.52 |
| PSO | CNN | 11.6 | 6.8 | 0.78 | 3.8 | 2.6 | 0.91 | 5.0 | 6.7 | 0.85 |
| | LSTM | 10.9 | 6.2 | 0.73 | 4.1 | 2.4 | 0.89 | 5.9 | 6.8 | 0.82 |
| | BERT | 11.9 | 8.2 | 0.72 | - | - | - | 7.8 | 7.9 | 0.84 |
| | RoBERTa | 12.3 | 8.4 | 0.70 | - | - | - | 8.3 | 8.1 | 0.81 |
| Reinforce-Bug | CNN | 13.3 | 7.8 | 0.81 | 3.8 | 2.3 | 0.91 | 6.5 | 6.2 | 0.87 |
| | LSTM | 14.7 | 7.6 | 0.79 | 3.9 | 2.7 | 0.91 | 6.9 | 6.1 | 0.84 |
| | BERT | 16.5 | 9.1 | 0.77 | 4.7 | 3.9 | 0.85 | 7.9 | 7.4 | 0.80 |
| | RoBERTa | 17.3 | 9.3 | 0.75 | 5.1 | 4.1 | 0.85 | 8.0 | 7.5 | 0.82 |
| BEAT | CNN | 15.3 | 7.3 | 0.67 | 3.8 | 1.9 | 0.89 | 6.3 | 6.1 | 0.63 |
| | LSTM | 13.4 | 7.1 | 0.65 | 3.7 | 2.3 | 0.88 | 6.6 | 6.0 | 0.54 |
| | BERT | 15.8 | 8.4 | 0.63 | 4.5 | 2.7 | 0.84 | 8.8 | 7.2 | 0.56 |
| | RoBERTa | 15.5 | 8.5 | 0.62 | 4.3 | 2.8 | 0.85 | 9.8 | 7.3 | 0.52 |
| BBA | CNN | 13.3 | 10.3 | 0.69 | 5.4 | 1.8 | 0.85 | 6.2 | 7.9 | 0.64 |
| | LSTM | 13.4 | 9.8 | 0.67 | 5.8 | 2.4 | 0.86 | 5.9 | 7.7 | 0.61 |
| | BERT | 14.5 | 10.9 | 0.63 | 6.5 | 2.7 | 0.81 | 7.3 | 9.7 | 0.57 |
| | RoBERTa | 14.9 | 11.3 | 0.61 | 6.3 | 2.7 | 0.81 | 7.6 | 10.2 | 0.53 |
| BESA | CNN | 12.3 | 9.8 | 0.85 | 2.3 | 2.2 | 0.93 | 4.5 | 7.2 | 0.87 |
| | LSTM | 10.4 | 9.6 | 0.87 | 2.1 | 1.9 | 0.92 | 5.3 | 7.3 | 0.86 |
| | BERT | 11.3 | 11.5 | 0.82 | 3.3 | 2.9 | 0.91 | 6.2 | 8.8 | 0.81 |
| | RoBERTa | 11.5 | 10.9 | 0.81 | 3.2 | 2.9 | 0.90 | 6.9 | 9.3 | 0.82 |
| ATGSL-SA | CNN | 11.3 | 9.6 | 0.88 | 2.0 | 2.1 | 0.96 | 3.8 | 7.4 | 0.92 |
| | LSTM | 10.5 | 9.4 | 0.91 | 1.9 | 2.2 | 0.96 | 4.2 | 7.6 | 0.89 |
| | BERT | 11.3 | 10.6 | 0.85 | 2.8 | 3.1 | 0.97 | 5.3 | 8.6 | 0.84 |
| | RoBERTa | 12.3 | 11.4 | 0.83 | 2.7 | 3.2 | 0.96 | 5.4 | 9.2 | 0.83 |
| ATGSL-BM | CNN | 12.5 | 8.3 | 0.82 | 3.4 | 1.3 | 0.92 | 5.4 | 3.2 | 0.81 |
| | LSTM | 12.3 | 7.3 | 0.84 | 3.2 | 1.2 | 0.91 | 5.3 | 3.0 | 0.83 |
| | BERT | 14.5 | 8.4 | 0.79 | 3.5 | 1.7 | 0.90 | 6.8 | 4.1 | 0.78 |
| | RoBERTa | 14.9 | 9.2 | 0.77 | 3.4 | 1.8 | 0.91 | 6.7 | 4.2 | 0.75 |
| ATGSL-FUSION | CNN | 9.6 | 9.2 | 0.89 | 3.4 | 2.3 | 0.93 | 3.1 | 6.9 | 0.88 |
| | LSTM | 9.4 | 9.5 | 0.88 | 1.7 | 2.5 | 0.94 | 3.2 | 6.8 | 0.86 |
| | BERT | 10.7 | 11.2 | 0.83 | 2.9 | 3.1 | 0.92 | 4.4 | 8.1 | 0.82 |
| | RoBERTa | 10.2 | 11.7 | 0.82 | 3.1 | 3.3 | 0.91 | 4.8 | 8.4 | 0.83 |

Table 4: Time (in seconds) needed in attacking the BERT.

| Dataset | PWWS | TFEO | PSO | Reinforce-Bug | BBA |
|---|---|---|---|---|---|
| MR | 6424 | 3830 | 4532 | 3890 | 5230 |
| IMDB | 18371 | 9532 | - | 6883 | 9872 |
| AG's News | 7857 | 8850 | 18531 | 9352 | 7533 |
| MNLI | 3171 | 2241 | 4642 | 2327 | 2581 |
| SNLI | 1871 | 941 | 3842 | 1037 | 1540 |

| Dataset | BEAT | BESA | ATGSL-SA | ATGSL-BM | ATGSL-FUSION |
|---|---|---|---|---|---|
| MR | 4328 | 7641 | 8785 | 4032 | 8327 |
| IMDB | 19371 | 56532 | 54132 | 8783 | 19872 |
| AG's News | 9691 | 17543 | 16543 | 8583 | 15231 |
| MNLI | 2751 | 3450 | 4392 | 2232 | 3573 |
| SNLI | 1658 | 2690 | 3213 | 1537 | 3542 |

replacement words due to the semantic uncertainty of generated words and lower semantic similarity.

Learning model-based attacks such as Reinforce-Bug and BBA show that using evaluation history to train the model has advantages in sentence quality and efficiency. ReinforceBug considers the text quality evaluation score in the reward to generate higher quality adversarial texts but cannot guarantee a high attack success rate due to its high variance. BBA reduces time cost but has difficulty in fitting complex target models using evaluation history (BERT, RoBERTa).

Our proposed attack algorithms have shown excellent performance in various aspects. Compared to BESA, which also uses the SA algorithm, our ATGSL-SA algorithm has a higher attack success rate and semantic similarity due to its ability to utilize a well-organized linguistic thesaurus and solve the problem of being trapped in local optimal solutions by using acceptance probability $p$ and variable temperature $T$. We also evaluate semantic consistency and grammar errors to ensure that readers do not change their initial predictions and maintain reading fluency. Our ATGSL-BM algorithm uses a well-trained conditional generative model to smooth out noise in the heuristic-defined search target and generate high-quality adversarial texts with lower attack costs. Additionally, ATGSL-FUSION has the highest attack success rate since it has the acceptance probability to use the adversarial texts generated by ATGSL-BM as the intermediate solution to avoid being trapped by initial conditions in local optima.

**Ablation study.** In addition, the bottom three rows of Tables 3 and 5 show the effects of ATGSL-SA and ATGSL-BM. These results demonstrate that our algorithms play important roles in semantic similarity and the quality of generated texts, respectively. We also conduct supplementary experiments with HowNet (ATGSL-SA without H), BERT-base (ATGSL-BM without fine-tuning), and other variants to analyze their performance in attack success rate, semantic similarity, runtime, and qrs. It is noticeable that the adversarial samples generated by ATGSL-SA and ATGSL-FUSION have higher semantic similarity, while those generated by ATGSL-BM have fewer grammar errors and higher fluency. During the attack phase, the time consumption and queries of ATGSL-BM are less than other variants. Compared to ATGSL-BM without fine-tuning, our ATGSL-BM has a higher ASR and semantic similarity. Additionally, we investigate how the amount of training data affects the ASR of ATGSL-BM on different datasets. As a result, short text datasets (e.g., MR, SNLI, MNLI) require less data to achieve high ASR than long text datasets (e.g., IMDB, AG's News). Please refer to Fig. 2 in Appendix D.2 for more details.

## 5.3 Human Evaluation

To validate and assess the quality of adversarial samples, we randomly sample 200 adversarial examples targeting LSTM on the MR dataset and targeting BERT on the SNLI dataset. Human judges were asked to evaluate the text similarity and gram-

Table 5: The analysis for all variants to attack the BERT model on MR. Qrs is the average number of queries.

| Dataset | Methods | ASR | %S | Time | Qrs |
|---|---|---|---|---|---|
| | ATGSL-SA (w/o $\mathbb{H}$) | 89.6% | **0.89** | 7854 | 79 |
| | ATGSL-SA | 94.1% | 0.85 | 8785 | 72 |
| MR | ATGSL-BM (w/o fine-tune) | 86.3% | 0.63 | 4848 | 63 |
| | ATGSL-BM (training process) ATGSL-BM (attack process) | 97.4% | 0.79 | 19652 **4032** | 55 **43** |
| | ATGSL-FUSION | **98.9%** | 0.83 | 8327 | 87 |

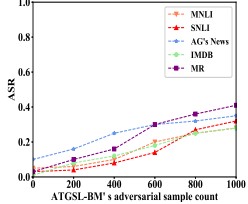 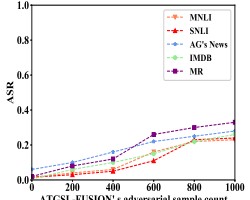

Figure 1: As more adversarial samples increase, the improvement in accuracy after ATGSL-SA and ATGSL-FUSION attacks is demonstrated.

matical correctness of the adversarial text generated by our method. As shown in Table 10, our method achieves higher scores in both text similarity and grammatical correctness. For more analysis of the results, please refer to Appendix D.4.

### 5.4 Adversarial Training

For adversarial training, we generate adversarial examples using 10% of samples from the IMDB and SNLI datasets' training sets. We then combine the generated adversarial examples with the original training sets of our respective datasets and retrain BERT. We then use our attack strategy to attack BERT again. Results are shown in Fig. 1. After the attack, the accuracy rate is increased by 15% to 35%. This indicates that adding adversarial samples to the training data makes the target model more robust to attacks.

### 5.5 Transferability

If an adversarial example is generated for a specific target model but also successfully attacks other target models, it is called a transferable example. We evaluate the transferability of adversarial attacks generated on the Ag's News dataset (Kurakin et al., 2018). The results are shown in Table 6. In Transfer-1, compared to previous attacks, our attack produces adversarial examples with a higher attack success rate, demonstrating better transferability. In addition, we conduct our experiments on the same binary sentiment classification datasets (MR, IMDB) in Transfer-2. Our ATGSL-BM still

Table 6: Transfer-1: The ASR of transferred adversarial examples on the AG's New. Transfer-2: Transfer results of ATGSL-BM model with the same emotion classification task. Higher ASR reflects higher transferability.

| Transfer-1 | PSO | BEAT | BESA | ATGSL-FUSION |
|---|---|---|---|---|
| CNN→BERT | 68.5% | 72.3% | 74.9% | **80.5%** |
| BERT→CNN | 72.4% | 75.5% | 76.3% | **84.4%** |
| Transfer-2 | CNN | LSTM | BERT | RoBERTa |
| IMDB→MR | 92.3% | 89.5% | 85.4% | 84.7% |
| MR→IMDB | 90.8% | 87.7% | 84.4% | 85.3% |

maintains a high attack success rate, demonstrating that our attack algorithm using fine-tuned pre-trained models has strong cross-dataset transferability.

## 6 Case Study

Tables 13 and 14 in Appendix D.5 show examples generated by BERT-ATTACK, PSO, ATGSL-SA, ATGSL-BM, and ATGSL-FUSION on the IMDB and SNLI datasets. The results indicate that our approaches exhibits better performance in terms of attack efficiency and text quality. For more analysis of the results, please refer to Appendix D.5.

## 7 Conclusion

In this paper, we introduced ATGSL, a new attack framework with three algorithms that balance attack efficiency and adversarial sample quality. ATGSL-SA used Simulated Annealing to search for similar adversarial texts. ATGSL-BM fine-tuned a pre-trained language model (BERT-MLM) to improve attack effectiveness and text quality. ATGSL-FUSION addressed the ATGSL-SA algorithm's susceptibility to initial conditions by using a trained language model to generate intermediate solutions. Our experiments show superior attack performance compared to baseline methods while maintaining a balance between attack performance and text quality.

### Limitations

Although we can see that ATGSL-BM has achieved a new state-of-the-art level and performed well on short text datasets (MR, SNLI, MNLI) while maintaining high attack efficiency at low cost, its performance on long text datasets (IMDB, AG's News) is not as good as ATGSL-SA. As shown in Figure 3, this is due to insufficient training data. Even if we can generate enough training samples from limited test data (multiple attacks), we cannot enrich

the variety of training samples. Our future work is to further learn more knowledge from successful or failed adversarial samples using self-supervised learning. On the other hand, the fine-tuning of the pre-trained text generator model used the typical BERT-MLM. On the other hand, the fine-tuning of the pre-trained text generator model used the typical BERT-MLM. In future work, we will continue to explore the expandability of our proposed attack framework by trying to integrate it with more popular pre-trained models, which is a key focus of our future work.

## Broader Ethical Impact

Our research focuses on the important problem of adversarial vulnerabilities of classification models on discrete sequential data. Even though it is possible that a malicious adversary misusing ATGSL to attack public text classification APIs, we believe our research can be a basis for the improvement in defending against adversarial attacks on discrete sequential data.

## ACKNOWLEDGEMENTS

This work is funded by the National Key R&D Program of China (2022YFB3103700, 2022YFB3103704)

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

## A  The Details of ATGSL-SA Algorithm.

The search process of ATGSL-SA can be divided into the following steps:

(1) At first, the algorithm randomly selects a replacement word and calculates a new $s(\mathbf{Y}_{true}|\mathbf{X}_{new})$.

(2) Judging based on Metropolis criteria: when $s(\mathbf{Y}_{ture}|\mathbf{X}_{new}) < s(\mathbf{Y}_{true}|\mathbf{X})$, update $\mathbf{X}$ to $\mathbf{X}_{new}$. When $s(\mathbf{Y}_{true}|\mathbf{X}_{new}) > s(\mathbf{Y}_{true}|\mathbf{X}_{adv})$, calculate the probability $p$ in the line 14 of Algorithm 1:

$$p' = e^{-(s(\mathbf{Y}_{true}|\mathbf{X}_{new})-s(\mathbf{Y}_{true}|\mathbf{X}_{adv}))/T_0}, \quad (9)$$

take a random number $r$ ($0 < r < 1$), update the $\mathbf{X}_{adv}$ to the $\mathbf{X}_{new}$ when $r < p$. It is not difficult to find that the probability of accepting $\mathbf{X}_{new}$ increases as temperature increases. After modification, if the classifier $\mathbf{F}$ is misled, we obtain the successfully attacked adversarial sample.

(3) Conduct a cooling:

$$T_0' = T_{init} - \frac{t}{T}(Sem(\mathbf{X}_{adv}-Sem(\mathbf{X}_{new}))) - C \times t. \quad (10)$$

If the $\mathbf{X}_{new}$ semantic similarity score is higher, then temperature increases. This design can prevent the SA algorithm from entering the optimal local solution early. Then, turn to step 1, repeat multiple times, and the result tends to be stabilized.

## B  Details of ATGS-BM Training Process

Fig. 2 and Algorithm 2 illustrates how the condition generator of ATGSL-BM learns from the search results of ATGSL-SA.

## C  Experiment Implementation Details

### C.1  Datasets

To show the wide applicability of ATGSL, we evaluate ATGSL on various datasets for classification tasks textual entailment.

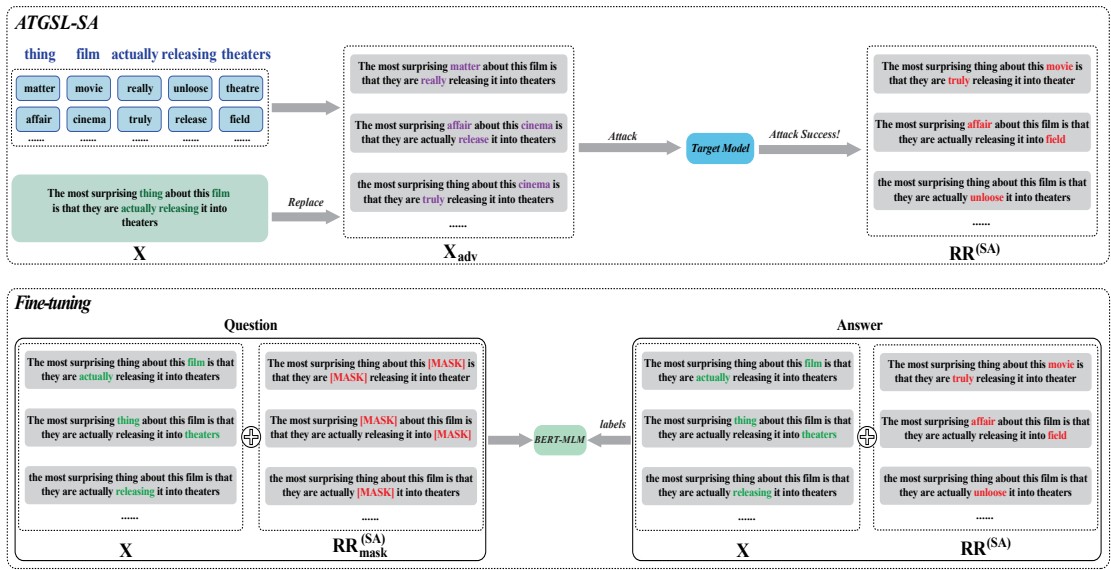

Figure 2: The diagram of ATGS-BM training process.

**Algorithm 2:** The Trained BERT-MLM

**Input:** Sample text $\mathbf{X}$, target model $\mathbf{F}$
**Output:** A fine-tuned BERT-MLM model

1  $\mathbf{RR}^{(SA)} \leftarrow \{\}$; // Pseudo-reference set is initially an empty set;
// Generate adversarial samples as pseudo-reference for training;

2  **for** *an input* $x_i \subset \mathbf{X}$ **do**

3      $s_i^{(SA)} = \text{ATGSL-SA}(x_i, x_i, \mathbf{C}_i, \mathbf{F})$;
// ATGSL-SA is detailed in Algorithm 1;

4      $\mathbf{RR}^{(SA)} \leftarrow \mathbf{RR}^{(SA)} \cup s_i^{(SA)}$;

5  **for** *all epochs* **do**

6      Fine-tune BERT-MLM by cross-entropy loss with pseudo-reference set $\mathbf{RR}^{(SA)}$ and its masked text $\mathbf{RR}_{mask}^{(SA)}$, conditioned on its origin text $\mathbf{X}$;

**Return:** Resulting the well-trained BERT-MLM

- AG's News (Zhang et al., 2015): A sentence-level dataset for classifying news-type sentences into 4 topics: World, Sports, Business, and Science.

- Movie Reviews (Pang and Lee, 2005): A sentence-level sentiment classification dataset composed of positive and negative movie reviews from Rotten Tomatoes.

- IMDB Polarity (Maas et al., 2011): A document-level dataset for binary sentiment classification composed of polar movie reviews from IMDB.

- MNLI matched (Williams et al., 2017): A textual entailment dataset composed of sentence pairs from transcribed speeches, popular fiction, and government reports. The task is to determine the relationship between a pair of concepts, premises, and hypotheses. The test set and training set are derived from the same sources.

- SNLI (Bowman et al., 2015): A dataset composed of 570K sentence pairs derived from image captions. The task is to determine the relationship between two sentences: whether the second sentence can be derived from the first sentence's implication, contradiction, or neutral relationship.

## C.2 Baselines

We compare the performance of ATGSL against the state-of-the-art methods as follow: (1) **PWWS** (Ren et al., 2019): A method chooses candidate words from WordNet and sorts word attack order by multiplying the word saliency and probability variation. (2) **TextFooler** (TEFO) (Jin et al., 2020): A method obtains synonyms close to the original word from the Glove space, and the word selection priority is determined by iteratively deleting input words and calculating the DNNs score changes. (3) **PSO** (Zang et al., 2020b): A method selects word candidates from HowNet and employs the PSO to find adversarial texts. (4) **ReinforceBug** (Sabir et al., 2021): A reinforced model that directly utilizes the prediction confidence score of the adversarial text in the target model and the sentence quality score as feedback to optimize the ef-

fect of the attack. (5) **BERT-ATTACK** (BEAT) (Li et al., 2020b): A method utilizes the BERT-MLM to generate candidate words and attack words in descending order with the static word importance score. (6) **BESA** (Yang et al., 2021b): A method leverages the BERT to generate candidate words and employs Simulated Annealing (SA) to determine the word substitution order adaptively. (7) **BBA** (Lee et al., 2022): A Bayesian optimized black-box attack method that dynamically calculates replacement positions based on query history dynamics and automatic correlation determination (ARD) classification rules.

We implement PWWS, PSO, TEFO and BEAT models using Open Source Framework (Zeng et al., 2021), and ReinforceBug, BESA, BBA and our method with Pytorch. To make a fair comparison, we set the upper bound of the number of replacing words as $M = 20$. Our method gives the parameter settings in line 1 of Algorithm 1 and Algorithm 2. In ATGSL-BM, we set $\{t, k\} = \{2, 10\}$. Moreover, in each dataset, we randomly select 5k test samples for multiple iterations to craft 40k attack successful adversarial samples for the training process. Finally, 1k test samples that the model has not seen before were selected as test data for the attack experiment.

# D Additional Experiment Results

## D.1 Original Accuracy of Various Datasets.

For each dataset, we train four state-of-the-art models on the training set and obtain test set accuracy scores similar to the original implementations, as shown in Table 7.

Table 7: Test accuracy of five datasets before attacks.

| Model | MR | IMDB | AG's News | Model | MNLI | SNLI |
|---|---|---|---|---|---|---|
| CNN | 78.3% | 83.2% | 90.9% | InferSent | 70.6% | 84.3% |
| LSTM | 79.3% | 84.5% | 89.3% | ESIM | 78.3% | 85.6% |
| BERT | 86.5% | 92.3% | 93.3% | BERT | 84.4% | 88.1% |
| Roberta | 87.1% | 93.5% | 94.1% | RoBERTa | 86.7% | 89.5% |

## D.2 Effect of Training Amount

As shown in Fig. 3, we found that short text datasets (e.g., MR, SNLI, MNLI) require less data to achieve high ASR, while long text datasets (e.g., IMDB, AG's News) need more data for the same purpose.

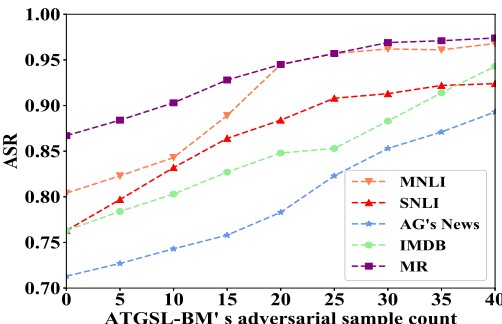

Figure 3: Effect of different amounts of training data on the ATGSL-BM.

## D.3 Character-level adversarial text generation

While our method is based on word-level adversarial text generation techniques, it's worth noting that our search-based learning framework is equally applicable to character-level adversarial text generation. Hence, we have incorporated comparative experiments involving character-level adversarial text generation. The baselines we refer to are:

- DeepWordBug(Gao et al., 2018): A character-level adversarial attack algorithm based on the differential evolution technique, introducing subtle character-level perturbations to generate adversarial text that leads to erroneous outputs from natural language processing models.

- PWWS(Ren et al., 2019): An attack algorithm utilizing white-box strategy, generating adversarial text through subtle character replacements, insertions, and deletions to mislead the classification output of natural language processing models.

Table 8: Classification accuracy on disturbed datasets using different attack methods. The third column represents the classification accuracy of the model for the original samples. Lower classification accuracy corresponds to more effective attack methods.

| Datasets | Model | Original | DeepWordBug | PWWS | ATGSL-SA | ATGSL-BM | ATGSL-FUSION |
|---|---|---|---|---|---|---|---|
| MR | CharCNN | 77.9% | 27.8% | 25.4% | 20.8% | 19.7% | 17.8% |
| | LSTM | 77.3% | 28.6% | 25.2% | 21.7% | 18.9% | 17.2% |
| | BERT | 86.5% | 38.3% | 30.2% | 23.8% | 21.3% | 19.3% |
| | RoBERTa | 87.1% | 37.8% | 31.6% | 23.1% | 22.3% | 19.6% |
| Ag's News | CharCNN | 89.3% | 32.8% | 25.8% | 23.1% | 22.5% | 16.8% |
| | LSTM | 89.3% | 35.6% | 23.9% | 22.8% | 20.4% | 18.9% |
| | BERT | 93.5% | 41.3% | 32.7% | 25.8% | 23.5% | 22.1% |
| | RoBERTa | 94.1% | 43.8% | 33.3% | 25.1% | 23.1% | 22.9% |

From Table 8 and 9, it is evident that at the character-level, our model consistently exhibits re-

Table 9: Word replacement rate of each attacking method on the selected models.

| Datasets | Model | DeepWordBug | PWWS | ATGSL-SA | ATGSL-BM | ATGSL-FUSION |
|---|---|---|---|---|---|---|
| MR | CharCNN | 18.8% | 13.4% | 8.1% | 7.7% | 7.3% |
| | LSTM | 17.3% | 14.2% | 8.4% | 6.9% | 7.5% |
| | BERT | 22.1% | 20.7% | 14.7% | 11.4% | 9.3% |
| | RoBERTa | 22.4% | 19.8% | 15.3% | 12.8% | 9.5% |
| Ag's News | CharCNN | 22.8% | 19.4% | 18.3% | 15.7% | 17.8% |
| | LSTM | 21.6% | 20.2% | 18.7% | 16.3% | 17.2% |
| | BERT | 27.3% | 26.7% | 24.7% | 22.4% | 19.3% |
| | RoBERTa | 26.5% | 26.8% | 23.6% | 22.8% | 18.2% |

Table 10: Human-Evaluation Results.

| Dataset | Method | Semantic | Grammar |
|---|---|---|---|
| MR | ATGSL-SA | **0.96** | 4.23 |
| | ATGSL-BM | 0.94 | **4.68** |
| | ATGSL-FUSION | 0.93 | 4.38 |
| SNLI | ATGSL-SA | **0.91** | 4.43 |
| | ATGSL-BM | 0.88 | **4.78** |
| | ATGSL-FUSION | 0.85 | 4.57 |

markable attack efficiency and a reduced word replacement rate. This observation highlights the adaptability and flexibility of our generative strategy across diverse text granularities.

### D.4 Details of Human Evaluation

In order to evaluate the quality of adversarial examples, we randomly select 200 samples that target LSTM on the MR dataset and BERT on the SNLI dataset. The true class labels of these samples are kept hidden and evaluators are asked to classify them. We find that 95% of adversarial examples in MR and 94% in SNLI have the same classification label as their original samples. In addition, five graduate students majoring in linguistics are provisionally recruited to annotate all the users according to their expertise experience. Human judges are asked to score each adversarial example on grammatical correctness and semantic similarity with the original example. They are instructed to score each example from 1 to 5 based on grammatical correctness and assign a score of 0, 0.5, or 1 for semantic similarity, following the practice of (Gagnon-Marchand et al., 2019; Jin et al., 2020). The results are shown in Table 10. Clearly, human judges find that the adversarial texts generated by ATGSL-BM have higher grammatical correctness. Additionally, they find that ATGSL-SA and ATGSL-BM, which utilize linguistic thesaurus (e.g., WordNet and HowNet), have higher semantic similarity.

Additionally, in Table 11 we will augment the comparative results between ATGSL and the baseline methods within the context of human assessments, as well as emphasize the label-preserving proficiency of ATGSL.

For each adversarial example, we have solicited human evaluators to assign scores based on three distinct aspects: label correctness, syntactic accuracy, and semantic similarity when compared to the original exemplar. The abbreviation "Acc" indicates the conformity of the adversarial sample to its original classification. Additionally, grammatical correctness score is evaluated on a scale from 1 to 5, where:

- Level 1: Text contains severe grammatical errors, rendering comprehension difficult.

- Level 2: Text exhibits multiple grammatical errors, impacting understanding.

- Level 3: Text features minor grammatical errors, but remains intelligible overall.

- Level 4: Text is essentially devoid of grammatical errors, flowing smoothly and comprehensibly.

- Level 5: Text is devoid of conspicuous grammatical errors, demonstrating exceptionally high grammatical precision.

Furthermore, semantic similarity is gauged through the allocation of scores, with values of 0, 0.5, and 1, as follows:

- 0: Generated text diverges significantly from the original text, with minimal shared semantics or themes.

- 0.5: Some semblance of similarity exists between the generated text and the original text, yet notable differences persist.

- 1: The generated text closely mirrors the original text, exhibiting a high degree of semantic and thematic consistency.

As shown in Table 11, it is evident that human evaluators have observed a higher degree of syntactic accuracy in the adversarial texts generated by our approach, particularly ATGSL-BM, compared to alternative methods (PWWS, PSO). Additionally, the utilization of linguistic lexicons such as WordNet and HowNet in ATGSL-SA has been found to result in enhanced semantic similarity.

Table 11: Human-Evaluation Results.

| Datasets | Method | Accuracy | Semantic | Grammar |
|---|---|---|---|---|
| | PWWS | 0.79 | 0.81 | 3.81 |
| | BEAT | 0.74 | 0.72 | 4.39 |
| MR | ATGSL-SA | 0.90 | **0.96** | 4.23 |
| | ATGSL-BM | 0.93 | 0.94 | **4.68** |
| | ATGSL-FUSION | **0.96** | 0.93 | 4.38 |
| | PWWS | 0.73 | 0.83 | 4.14 |
| | BEAT | 0.71 | 0.67 | 4.54 |
| SNLI | ATGSL-SA | 0.89 | **0.91** | 4.43 |
| | ATGSL-BM | 0.86 | 0.86 | **4.78** |
| | ATGSL-FUSION | **0.91** | **0.91** | 4.57 |

Table 12: The attack success rate (ASR) in high/low semantic similarity

| Datasets | Method | ASR(High Semantic) | ASR(Low Semantic) |
|---|---|---|---|
| | PWWS | 0.81 | 0.73 |
| | BEAT | 0.83 | 0.84 |
| MR | ATGSL-SA | 0.96 | 0.80 |
| | ATGSL-BM | 0.96 | 0.97 |
| | ATGSL-FUSION | 0.97 | 0.93 |
| | PWWS | 0.84 | 0.80 |
| | BEAT | 0.88 | 0.86 |
| SNLI | ATGSL-SA | 0.94 | 0.87 |
| | ATGSL-BM | 0.95 | 0.95 |
| | ATGSL-FUSION | 0.98 | 0.93 |

Moreover, our method demonstrates superior accuracy in correctly categorizing texts, highlighting its strengthened label-preserving capability.

We also provide the attack success rate (ASR) under different similarities and label consistencies. We classify adversarial examples generated by our proposed method as follows: high similarity and human judgment of the same category (HS), high similarity and inconsistent label (HI), low similarity and human judgment of the same category (LS), low similarity and inconsistent labels (LI).

For our ATGSL-SA on the MR dataset, targeting the BERT model, we achieved an Attack Success Rate (ASR) of 94.1%, with a total of 814 successfully attacked samples. The distribution across our four categories is as follows: HS:HI:LS:LI = 773:32:24:6.

Taking the MR dataset and BERT as the target model as an example, the attack success rate (ASR) for ATGSL-SA in high-similarity adversarial examples is calculated as follows: ASR = HS / (HS + HI) = 773 / (773 + 32) ≈ 0.960. In low-similarity adversarial examples, the ASR is calculated as ASR = LS / (LS + LI) = 24 / (24 + 6) = 0.8.

For our ATGSL-BM on the MR dataset, targeting the BERT model, we achieved an ASR of 97.4%, with a total of 842 successfully attacked samples. The distribution across the four categories is as follows: HS:HI:LS:LI = 744:28:58:12. For ATGSL-BM, in high-similarity adversarial examples, the ASR is calculated as ASR = HS / (HS + HI) = 744 / (744 + 28) ≈ 0.963. In low-similarity adversarial examples, the ASR is calculated as ASR = LS / (LS + LI) = 24 / (24 + 6) ≈ 0.967.

These results illustrate the effectiveness of the adversarial attack methods under consideration, with slightly higher ASR values for ATGSL-BM in both high and low-similarity adversarial examples compared to ATGSL-SA.

As observed from the Table 12, our approach demonstrates outstanding performance in both high and low similarity categories. For heuristic-based algorithms like PWWS and ATGS-SA, they excel in generating adversarial samples with high similarity, primarily relying on synonym replacement. Consequently, the attack effectiveness is better in high similarity cases. This can be attributed to the inclusion of similarity scores in the objective and cooling functions of ATGS-SA. The objective is to slightly sacrifice similarity to achieve a broader search space in cases of continuous attack failures. This explains the drop in similarity in lower similarity categories, where ASR may decrease.

On the other hand, ATGS-BM and BEAT are language model-based methods and are less sensitive to the similarity of adversarial samples compared to heuristic algorithms. Therefore, they perform well in both categories. However, ATGS-BM generates a significantly higher number of high similarity adversarial samples compared to BEAT.

### D.5 Details of Case Study

Tables 13 and 14 show examples generated by BERT-ATTACK, PSO, ATGSL-SA, ATGSL-BM, and ATGSL-FUSION on the IMDB and SNLI datasets. Compared to ATGSL, although PSO searches for adversarial samples in synonym and semantic spaces, it does not consider the text similarity generated during the iteration process, resulting in lower adversarial text similarity. In contrast, the ATGSL algorithm has a certain probability of accepting suboptimal solutions, making it easier to find the optimal solution. Additionally, ATGSL-BM has fewer syntax errors and higher attack efficiency than algorithms generated by heuristic methods. Compared to BERT-ATTACK, which is also based on pre-trained models, ATGSL-BM has higher text similarity and ASR. Furthermore, ATGSL-FUSION is relatively stable and maintains

a good balance between attack efficiency and ad-
versarial text quality.

Table 13: Adversarial examples by attacking BERT model on MR dataset.

| | |
|---|---|
| BEAT-ATTACK (Successful attack. True label score = 24.91%, semantic similarity score = 0.36, qrs=164, grammaticality score = 2) | An ~~incontrovertible~~ ~~theontroveudibility~~ **contemporary** french ~~psychological~~ **grief** ~~drama examining~~ **tragedy** the ~~standoff~~ **relationship** of an aloof father and his freeze son after 20 years apart. |
| PSO (Successful attack. True label score = 33.26%, semantic similarity score = 0.56, qrs=150, grammaticality score = 3) | An incontrovertible french psychological ~~drama~~ **dramatic** ~~examining~~ **analyse** the standoff of an aloof ~~father~~ **begetter** and his ~~freeze~~ **freezing** son after 20 years ~~apart~~ **aside**. |
| ATGSL-SA (Successful attack. True label score = 44.86%, semantic similarity score = 0.75, qrs=84, grammaticality score = 2) | An ~~incontrovertible~~ **inarguable** french ~~psychological~~ **unworldly** drama examining the standoff of an aloof ~~father~~ **begetter** and his freeze ~~son~~ **boy** after 20 years ~~apart~~ **asunder**. |
| ATGSL-BM (Successful attack. True label score = 44.86%, semantic similarity score = 0.84, qrs = 56, grammaticality score = 1) | An incontrovertible ~~french~~ **russian** ~~psychological~~ **psychiatric** drama ~~examining~~ **question** the standoff of an aloof father and his freeze son after 20 years apart. |
| ATGSL-FUSION (Successful attack. True label score = 31.28%, semantic similarity score = 0.91, qrs = 94, grammaticality score = 2) | An incontrovertible french psychological ~~drama~~ **seriocomedy** examining the standoff of an aloof father and his ~~freeze~~ **trammel** son after 20 years apart. |

Table 14: Adversarial examples by attacking BERT model on SNLI dataset.

| |
|---|
| **Premise**:A young woman with brown hair and an elderly man with gray hair and a sweater jump in the air on a snowy hill with snowshoes on their feet. |
| An ~~old~~ **dead** ~~woman~~ **women** and a young man are crossing the street |
| **Method**: BERT-ATTACK (Successful attack. True label score = 19.78%, qrs = 43, semantic similarity score = 0.33, grammaticality score = 1) |
| An ~~old~~ **abandoned** woman and a young man are ~~crossing~~ **frustrate** the ~~street~~ **route**. |
| **Method**: PSO (Successful attack. True label score = 22.28%, semantic similarity score = 0.54, qrs = 68, grammaticality score = 2) |
| An old ~~woman~~ **female** and a young ~~man~~ **brother** are crossing the street. |
| **Method**: ATGSL-SA (Successful attack. True label score = 24.67%, semantic similarity score = 0.73, qrs = 43, grammaticality score = 1) |
| An old woman and a young man are crossing the ~~street~~ **trajectory** |
| **Method**: ATGSL-BM (Successful attack. True label score = 28.78%, semantic similarity score = 0.88, qrs = 17, grammaticality score = 1) |
| An old woman and a ~~young~~ **tender** ~~man~~ **husband** are crossing the street |
| **Method**: ATGSL-FUSION (Successful attack. True label score = 20.58%, semantic similarity score = 0.79, qrs = 52, grammaticality score = 1) |