# OpenReview forum: "Adversarial Text Generation by Search and Learning"
_EMNLP/2023/Conference — EMNLP 2023 Findings_

### Official Review · Reviewer_xqBr · 2023-08-02

**Soundness:** 3

**Excitement:**

3: Ambivalent: It has merits (e.g., it reports state-of-the-art results, the idea is nice), but there are key weaknesses (e.g., it describes incremental work), and it can significantly benefit from another round of revision. However, I won't object to accepting it if my co-reviewers champion it.

**Paper Topic And Main Contributions:**

This paper combines heuristic search and a fine-tuned language model to generate adversarial attacks on black-box classifiers of natural language input. Search is performed over synonyms and sememes of words that appear in the input sentences and the replacement words are also used to fine-tune a BERT model to produce semantically similar words. The heuristic search and the model can be used separately or combined into a stronger attack.

**Questions For The Authors:**

Question a: You say you're using GPT-2 to create a grammar score. How are you calculating a grammar score based on the outputs of GPT? I would assume some kind of perplexity or sequence probability. Is it normalized to sequence length?

Question b: The output values for Sem() and Gram() aren't clear and without knowing what good scores and bad scores are and what their ranges are, it's hard for me to understand why you would want to divide semantic similarity by grammaticality. Could you maybe plot your score over the range of values they can take and show the cutoff value $\epsilon$?

Question c: Are all attacks using the same sem()/gram() score as their constraint? Results vary greatly depending on the constraints and if attacks use tighter or looser constraints that could put them at a (dis-)advantage. Also, do you know how many examples are left after filtering them out according to the constraints? If there's only one possible replacement then search may not be doing much.

Question d: Section 3 and equation 2 seem to suggest that you're using hard constraints to filter out examples that don't have a score better than $\epsilon$. Line 11 in Algorithm 1 and equation 5 seem to suggest that this is a soft constraint modulated by $\alpha$ and $\beta$, which is correct?

Question e: Why are you measuring attacks in terms of time to attack instead of queries? If you're querying a black-box model by API, you pay per number of API calls. Previous work has also measured queries as it shows how many examples actually need to be tried out. With learned attacks, faster times could be a result of different model choices and not fewer queries.

**Reasons To Accept:**

The method in this paper appears to improve upon other previous adversarial attacks by combining the strengths of both a search based method and a learned model based method.

**Reasons To Reject:**

There are multiple inconsistencies throughout the paper on their methodology and what is actually implemented. One section seems to suggest that hard constraints are being used to filter examples but another section suggests that soft constraints are being used.

I'm also not sure if the comparisons are fair due to the uncertainty over the constraints. Changing the constraints can wildly affect the ability of an attack method to find replacements or if there even replacements at all.

There are some inaccuracies such as referring to BERT as an encoder-decoder model and confusion from variables gaining/losing hats, primes, and super/subscripts.

**Reproducibility:**

2: Would be hard pressed to reproduce the results. The contribution depends on data that are simply not available outside the author's institution or consortium; not enough details are provided.

**Reviewer Confidence:**

2: Willing to defend my evaluation, but it is fairly likely that I missed some details, didn't understand some central points, or can't be sure about the novelty of the work.

**Typos Grammar Style And Presentation Improvements:**

Algorithm 1 doesn't use $\mathcal{B}$ anywhere in it. Addititionally, line 14 calculates $p'$ but the next few lines use $p$ and $1-p$.

Starting on line 462 you write: "We compare our method with these baselines such as PWWS, TEFO, PSO, ReinforceBug, BEAT, BESA, the PPWS, the PWWS, TEFO, BEAT are rule based attacks and the PSO, BESA and PWWS are Learning-based attacks". PWWS is repeated four times here and it's included in both learning and rule based attacks.

---

> ### Author Rebuttal · Authors · 2023-08-27
>
> I deeply apologize for any inconvenience that the formulaic expressions in the problem description might cause you. We greatly appreciate your understanding. We will continue to address your inquiries and subsequently revise the text concisely, aiming to enhance your reading efficiency.
>
> ### Answer a:
> I understand your perspective. Indeed, the accurate situation is that we used LanguageTool(https://languagetool.org) to identify grammatical errors, rather than GPT-2, due to an oversight on my part. We deeply apologize for any inconvenience caused by this oversight. Please rest assured, we are committed to rectifying this issue and making the necessary corrections.
>
> ### Answer b:
> Taking into consideration the potential for reader misunderstanding stemming from the presence of $Gram()/Sem()$ terms within the objective function, we are inclined to implement certain revisions. To elucidate, our primary objective lies in fostering the generation of high-quality adversarial texts by the model while simultaneously ensuring maximal preservation of textual coherence and fluency. As a matter of fact, we have not integrated stringent constraints such as $Gram()/Sem()$ into the model algorithm.To alleviate any potential misconceptions, in accordance with your feedback, we intend to make the following amendments to the final version of the problem description:
>
> This attack can generally be modelled as an optimization problem, which aims to mislead the target model by the adversarial sample $\mathbf{X}_{adv}$ with better quality of adversarial texts:
>
>
> $\mathbf{F}(\mathbf{X}_{adv})\neq \mathbf{Y}$,
>
>
> An adversarial text example $\mathbf{X}_{adv}$ is defined as the original input $\mathbf{X}$ that has been subjected to slight perturbations $\Delta \mathbf{X}$, i.e.,
>
>
> $\mathbf{X}_{adv} = \mathbf{X} + \Delta \mathbf{X}$ with the overarching goal of preserving a notable degree of semantic similarity and syntactic fluency.
>
> ### Answer c:
> I understand your concerns. This issue should be similar to the previous one. In reality, our algorithmic procedure does not incorporate a stringent filtering mechanism based on $Sem()/Gram()$. Our ATGSL-SA method, akin to the approach adopted by Simulated Annealing (BESA as in [1]), indeed integrates the semantic score within the objective function formulation for the adversarial text generation search process (Eq. 11), utilizing a and b as balance parameters. However, the Gram score is not employed. We express our apologies once again.
>
> ### Answer d:
> The present inquiry bears resemblance to the preceding one. Our approach involves the utilization of the Sem() as a constituent of the objective function design. The objective is to effectively explore adversarial text instances characterized by elevated attack potency and substantial textual similarity, akin to the approach outlined in Eq. 10 of BESA. Indeed, ATGSL-SA represents an enhanced iteration of BESA, incorporating refinements such as the integration of synonym tables and the introduction of a temperature-cooling function within the objective function. The latter adjustment serves to mitigate the influence of initial conditions and the potential entrapment in local optima. In addition, our core innovation lies in utilizing novel search and learning methods to enhance the attack efficiency and text quality of language models. Moreover, the widespread popularity of large pre-trained models also attests to the substantial benefits our model achieves in practical scenario applications far outweighing its training costs.
>
> ### Answer e:
> I appreciate your perspective. We have taken into full consideration the pivotal nature of both the average query count (Qrs) and the time cost as crucial metrics for model evaluation. It is noteworthy that certain studies, recognizing the inherent instability of Qrs due to factors such as uncertainties in model response, have accorded greater significance to time cost, as documented in references [1]~[3]. Conversely, while we consciously avoided direct Qrs comparisons within Tables 3 and 4 in our paper, it is imperative to underscore that in Experiment 5 of the ablation study, we indeed present Qrs metrics. Moreover, in the forthcoming final version, we intend to incorporate the Qrs comparison between ATGSL and alternative methods, as illustrated in the following Table 1. In contrast to other methodologies, our approach (especially ATGSL-BM) exhibits a proclivity for fewer Qrs, as substantiated by our supplementary experiments.
>
> #### Table 1: Qrs (average number of queries) needed in attacking the BERT.
>
> |Datasets| PWWS|TEFO|PSO|ReinforceBug|BBA|
> |:--------:| --------:| :--------: |:--------:|:--------:| :--------:|
> |MR|126|130|172|152|128|
> |SNLI|64|73|110|94|63|
>
> |Datasets| BEAT|BESA|ATGSL-SA|ATGSL-BM|ATGSL-FUSION|
> |:--------:| --------:| :--------: |:--------:|:--------:| :--------:|
> |MR|98|164|83|55|87|
> |SNLI|33|83|37|23|42|
>
> ### Answer f:
> Regarding the matter of reproducibility, multiple reviewers have noted the commendable reproducibility score of our work. In actuality, our model features a limited number of hyperparameters, with their respective values being delineated in Sections 4.1 (The values of $\alpha$ and $\beta$ are outcomes derived from repeated experimentation.) and 5.1. Furthermore, we have provided a comprehensive codebase, training procedures, and outcomes within the supplementary materials. The meticulously trained model parameters are also slated for public dissemination online. This comprehensive approach underscores the robust reproducibility inherent in our model architecture.
>
>
>
> ###  Notes:
> Thank you very much for your review and feedback. Multiple reviewers have acknowledged the uniqueness and significance of our proposed search/learning framework, utilizing search outcomes to enhance the attack performance and text quality of language models. Additionally, our approach has undergone comprehensive evaluation across multiple datasets and baselines. In response to the reviewers' feedback, we conducted supplementary experiments at various levels of text granularity and achieved significant results, further refining the assessment of label-preserving capabilities in human evaluations. This submission holds immense importance to me, as it provides a pivotal opportunity to showcase our extensive efforts and accomplishments. We earnestly request the reviewers to reconsider our work, and we express our heartfelt gratitude for your consideration.
>
>
> Reference:
>
> [1] Xinghao Yang, Weifeng Liu, Dacheng Tao, and Wei Liu. 2021b. Besa: Bert-based simulated annealing for adversarial text attacks. In International Joint Conference on Artificial Intelligence.
>
> [2] Xinghao Yang, Weifeng Liu, James Bailey, Dacheng Tao, and Wei Liu. 2021. Bigram and unigram based text attack via adaptive monotonic heuristic search. In Proceedings of the AAAI Conference on Artificial Intelligence, volume 35, pages 706–714.
>
> [3] Bushra Sabir, Muhammad Ali Babar, and Raj Gaire. 2021. Reinforcebug: A framework to generate adver- sarial textual examples. In Proceedings of the 2021 Conference of the North American Chapter of the Association for Computational Linguistics: Human Language Technologies, pages 5954–5964.re

---

### Official Review · Reviewer_9DV4 · 2023-08-04

**Soundness:** 4

**Excitement:**

3: Ambivalent: It has merits (e.g., it reports state-of-the-art results, the idea is nice), but there are key weaknesses (e.g., it describes incremental work), and it can significantly benefit from another round of revision. However, I won't object to accepting it if my co-reviewers champion it.

**Paper Topic And Main Contributions:**

This paper proposes a search and learning framework for Adversarial Text Generation by Search and Learning (ATGSL) and develops three novel black-box adversarial attack methods (ATGSL-SA, ATGSL-BM, ATGSL-FUSION). The core idea of ATGSL is regarding adversarial example generation as a unsupervised text generation problem, such that some search and learning modules can be adapted to craft high-quality adversarial examples.

**Questions For The Authors:**

A. Could ATGSL be applied to more granularities, like character-level and sentence-level?
B. Have you done some human evaluations regarding 1) comparing ATGSL to previous baselines; 2) the label-preserving ability of ATGSL?

**Reasons To Accept:**

1. The proposed ATGSL framework is novel and effective, which is proven by the given experimental results.
2. Experimental results show that ATGSL outperforms several black-box attack baselines on various classification datasets.

**Reasons To Reject:**

1. It seems that ATGSL is still a word-substitution based adversarial attack. Could ATGSL be applied to more granularities, like character-level and sentence-level?
2. Regarding the human evaluation section. Human evaluation is important when evaluating textual adversarial attacks since it is hard to infer whether the semantic meaning of an adversarial example has been changed by word substitutions from automatic evaluation metrics. However, the human evaluation section only provides the comparison between three variants of ATGSL. It also does not include the evaluation on label-preserving for ATGSL.

**Reproducibility:**

4: Could mostly reproduce the results, but there may be some variation because of sample variance or minor variations in their interpretation of the protocol or method.

**Reviewer Confidence:**

4: Quite sure. I tried to check the important points carefully. It's unlikely, though conceivable, that I missed something that should affect my ratings.

**Typos Grammar Style And Presentation Improvements:**

black box attack--balck-box attack

Line 45: neutral--neural

---

> ### Author Rebuttal · Authors · 2023-08-28
>
> ### Answer a:
>
> I comprehend your viewpoint. While our method is based on word-level adversarial text generation techniques, it's worth noting that our search-based learning framework is equally applicable to character-level adversarial text generation. Hence, we have incorporated comparative experiments involving character-level adversarial text generation.
>
> Baselines:
> DeepWordBug([1]): A character-level adversarial attack algorithm based on the differential evolution technique, introducing subtle character-level perturbations to generate adversarial text that leads to erroneous outputs from natural language processing models.
>
> PWWS([2]): An attack algorithm utilizing white-box strategy, generating adversarial text through subtle character replacements, insertions, and deletions to mislead the classification output of natural language processing models.
>
>
> #### Table 1: Classification accuracy on disturbed datasets using different attack methods. The third column represents the classification accuracy of the model for the original samples. Lower classification accuracy corresponds to more effective attack methods.
>
> | Datasets | Model | Original | DeepWordBug | PWWS | ATGSL-SA | ATGSL-BM | ATGSL-FUSION |
> | -------- |:--------:| --------:| :--------:| :--------:| --------:| :--------:| :--------:|
> |MR      |CharCNN |             77.9%|     27.8%        |        25.4%  |  20.8%     |  19.7%     |          17.8%     |
> |        |    LSTM       |     77.3%    | 28.6%      |          25.2% |   21.7%  |     18.9%      |          17.2%    |
> |      |      BERT        |              86.5%|     38.3%      |          30.2%     |23.8%     |  21.3%    |           19.3%|
> |      |      RoBERTa |               87.1% |    37.8%   |             31.6% |    23.1%  |     22.3%  |             19.6%|
> |Ag’s News|    CharCNN |            89.3%  |  32.8% |               25.8%  |  23.1%    |   22.5%            |     16.8% |
> ||             LSTM     |       89.3% |    35.6% |               23.9%|   22.8%  |     20.4% |               18.9% |
> ||BERT             |         93.5%| 41.3%    |            32.7%  |   25.8%   |    23.5%       |        22.1%|
> || RoBERTa     |           94.1% |43.8%   |             33.3%|     25.1%  |     23.1%     |          22.9%      |
>
>
>
>
>
>
>
> #### Table 2:  Word replacement rate of each attacking method on the selected models.
> |Datasets  |Model      |             DeepWordBug | PWWS| ATGSL-SA| ATGSL-BM |ATGSL-FUSION|
> | :--------: |:--------:| :--------:| -------- |:--------:| :--------:| -------- |
> |MR |     Char CNN    |             18.8%        |       13.4%  |  8.1%  |     7.7%   |            7.3%    |
> |      |        LSTM            |          17.3%       |        14.2%  |  8.4%    |   6.9%   |             7.5%   |
> ||              BERT            |           22.1%       |       20.7% |    14.7%   |    11.4%      |         9.3%|
> |  |            RoBERTa       |         22.4%         |      19.8%     | 15.3%   |   12.8%           |    9.5%|
> |Ag’s News|  Char CNN |         22.8% |              19.4% |   18.3%  |     15.7%     |          17.8%     |
> |   |          LSTM         |             21.6%    |           20.2%  |  18.7%    |   16.3%   |             17.2%|
> |  |            BERT     |                  27.3% |             26.7%   |  24.7%     |  22.4%        |       19.3%|
> | |         RoBERTa     |           26.5%         |      26.8%   |   23.6%     | 22.8%         |      18.2%|
>
> From Tables 1 and 2, it is evident that at the character-level, our model consistently exhibits remarkable attack efficiency and a reduced word replacement rate. This observation highlights the adaptability and flexibility of our generative strategy across diverse text granularities.
>
>
>
> ###  Answer b:
> Certainly, we intend to provide a detailed elucidation of the score distribution in our manual evaluation process. Additionally, we will augment the comparative results between ATGSL and the baseline methods within the context of human assessments, as well as emphasize the label-preserving proficiency of ATGSL.
>
> For each adversarial example, we have solicited human evaluators to assign scores based on three distinct aspects: label correctness, syntactic accuracy, and semantic similarity when compared to the original exemplar. The abbreviation "Acc" indicates the conformity of the adversarial sample to its original classification. Additionally, grammatical correctness score is evaluated on a scale from 1 to 5, where:
>
> Level 1: Text contains severe grammatical errors, rendering comprehension difficult.
> Level 2: Text exhibits multiple grammatical errors, impacting understanding.
> Level 3: Text features minor grammatical errors, but remains intelligible overall.
> Level 4: Text is essentially devoid of grammatical errors, flowing smoothly and comprehensibly.
> Level 5: Text is devoid of conspicuous grammatical errors, demonstrating exceptionally high grammatical precision.
> Furthermore, semantic similarity is gauged through the allocation of scores, with values of 0, 0.5, and 1, as follows:
>
> 0: Generated text diverges significantly from the original text, with minimal shared semantics or themes.
> 0.5: Some semblance of similarity exists between the generated text and the original text, yet notable differences persist.
> 1: The generated text closely mirrors the original text, exhibiting a high degree of semantic and thematic consistency.
>
>
> ####   Table 3: Human-Evaluation Results.
>
> |Dataset| Method |Accuracy| Semantic |Grammar|
> | :--------: |:--------:| --------:| :--------: |:--------:|
> |MR|       PWWS | 0.79    |    0.81    |   3.81|
>  ||             BEAT|    0.74  |      0.72   |    4.39|
> ||            ATGSL-SA |0.90 |   0.96  |      4.23|
> ||            ATGSL-BM| 0.93 |   0.94 |   4.68|
> ||            ATGSL-FUSION| 0.96| 0.93 |4.38   |
> |SNLI|       PWWS|  0. 73|       0.83  |     4.14|
> ||             BEAT   | 0.71  |      0.67     |  4.54|
> ||            ATGSL-SA |0.89   | 0.91  |    4.43|
> ||            ATGSL-BM |0.86|    0.86  |  4.78|
> ||         ATGSL-FUSION| 0.91 |0.91| 4.57|
>
>
> As shown in Table 3, it is evident that human evaluators have observed a higher degree of syntactic accuracy in the adversarial texts generated by our approach, particularly ATGSL-BM, compared to alternative methods (PWWS, PSO). Additionally, the utilization of linguistic lexicons such as WordNet and HowNet in ATGSL-SA has been found to result in enhanced semantic similarity. Moreover, our method demonstrates superior accuracy in correctly categorizing texts, highlighting its strengthened label-preserving capability.
>
> ###  Answer c:
> Regarding the innovation and limitations of our model, we leverage novel search and learning methods to enhance the attack efficiency and text quality of language models. It is worth noting that in the context of incremental experiments related to limitations, there are instances where ATGSL-BM's results do not surpass those of ATGSL-SA, which can be attributed to certain tricks employed within ATGSL-SA, such as the innovative cooling function. In essence, our ATGSL-BM significantly outperforms existing state-of-the-art methods, confirming the pronounced advantages and notable scalability of our approach. While further enhancing our ATGSL-BM through tricks is indeed feasible, such improvements could be pursued as a separate avenue of research. Additionally, the widespread popularity of large pre-trained models underscores that the substantial gains achieved in practical scenario applications far outweigh the costs associated with their training. We earnestly request reviewers to reconsider our work in light of these aspects.
>
> ###  Notes:
> I greatly appreciate your review and feedback. Multiple reviewers have recognized the uniqueness and significance of our proposed search/learning framework, leveraging pre-trained language models to enhance attack performance and text quality. Additionally, our approach has been meticulously evaluated across multiple datasets and baselines. In response to the reviewers' feedback, we conducted supplementary experiments and achieved significant results. This submission holds immense importance to me as it offers a pivotal opportunity to showcase our extensive efforts and achievements. We earnestly request the reviewers to reevaluate our work, and we express our heartfelt gratitude for your consideration.
>
> Reference:
> [1]Gao, J.; Lanchantin, J.; Soffa, M. L.; and Qi, Y. 2018. Black- box generation of adversarial text sequences to evade deep learning classifiers. In 2018 IEEE Security and Privacy Workshops (SPW), 50–56. IEEE.
> [2]Shuhuai Ren, Yihe Deng, Kun He, and Wanxiang Che. 2019. Generating natural language adversar- ial examples through probability weighted word saliency. In Proceedings of the 57th annual meeting of the association for computational linguistics, pages 1085–1097.

---

### Official Review · Reviewer_UFnh · 2023-08-11

**Typos Grammar Style And Presentation Improvements:** In line 412, it's said "we use use us…
**Soundness:** 4

**Excitement:**

4: Strong: This paper deepens the understanding of some phenomenon or lowers the barriers to an existing research direction.

**Missing References:**

Not that I know of.

**Paper Topic And Main Contributions:**

This paper introduces a novel attack framework named ATGSL, consisting of three components balancing attack performance and text quality. Specifically, it starts with the ATGSL-SA algorithm to efficiently search for replacement words. To improve iteration time cost, they authors propose ATGSL-BM and ATGSL-FUSION algorithms. While ATGSL-BM leverages a finetuned LM to find optimal search patterns,  ATGSL-FUSION uses search results of ATGSL-BM as intermediate solutions. The proposed framework outperforms existing methods in terms of both attack success rate and text quality maintenance.

Overall, this work’s contributions are in twofold: 1) their proposed algorithms are novel in a sense that it proposes search/learning methods and leverages pre-trained LM to improve attack performance/text quality; 2) their proposed algorithms outperform all baselines and successfully passed the evaluation from humans’ lens.


**Questions For The Authors:**

Q1. For ATGSL-SA, I wonder how hyper-parameter values (alpha/beta) are selected.
Q2. Yo mentioned grammar errors are evaluated via GPT-2. Why did you use GPT-2 when there are more modern models?


**Reasons To Accept:**

Here are several reasons to accept:
1. The paper contains a thorough literature review, which further highlights the novelty of their approach.
2. The paper’s method is unique in a sense that it proposes search/learning methods and leverages pre-trained LM to improve attack performance/text quality.
3. The authors evaluated their approach very thoroughly across multiple datasets and baselines.

**Reasons To Reject:**

One weakness I can think of is, given the limitations of training data size, pre-trained LM does not outperform the ATGSL-SA for long text datasets. It would've been nice if all results remained consistent regardless of the length of datasets.

**Reproducibility:**

5: Could easily reproduce the results.

**Reviewer Confidence:**

2: Willing to defend my evaluation, but it is fairly likely that I missed some details, didn't understand some central points, or can't be sure about the novelty of the work.

---

### Meta-Review · Area_Chair_3Mqk · 2023-09-16

**Recommendation:** 3

**Metareview:**

This paper treats black-box text attack as an unsupervised text generation problem and proposes a search and learning framework (ATGSL) with three adversarial attack methods (ATGSL-SA, ATGSL-BM, ATGSL-FUSION). ATGSL is significantly superior to the most advanced methods in terms of attack efficiency and adversarial text quality. On the other hand, the comparisons may not be fair due to uncertainty over the constraints.

---

### Decision · Program_Chairs · 2023-10-07

**Decision:**

Accept-Findings

**Comment:**

This paper treats black-box text attack as an unsupervised text generation problem and proposes a search and learning framework (ATGSL) with three adversarial attack methods (ATGSL-SA, ATGSL-BM, ATGSL-FUSION). ATGSL is significantly superior to the most advanced methods in terms of attack efficiency and adversarial text quality. On the other hand, the comparisons may not be fair due to uncertainty over the constraints.